# Y RNA: An Overview of Their Role as Potential Biomarkers and Molecular Targets in Human Cancers

**DOI:** 10.3390/cancers12051238

**Published:** 2020-05-14

**Authors:** Caterina Gulìa, Fabrizio Signore, Marco Gaffi, Silvia Gigli, Raffaella Votino, Roberto Nucciotti, Luca Bertacca, Simona Zaami, Alberto Baffa, Edoardo Santini, Alessandro Porrello, Roberto Piergentili

**Affiliations:** 1Department of Urology, Misericordia Hospital, 58100 Grosseto, Italy; 85cate@live.it (C.G.); roberto.nucciotti@uslsudest.toscana.it (R.N.); urologia.santini@gmail.com (E.S.); 2Department of Obstetrics and Gynecology, Misericordia Hospital, 58100 Grosseto, Italy; fabrizio.signore@uslsudest.toscana.it (F.S.); raffaella.votino@uslsudest.toscana.it (R.V.); alberto.baffa@uslsudest.toscana.it (A.B.); 3Pediatric Surgery and Urology Unit, Azienda Ospedaliera San Camillo-Forlanini, 00152 Rome, Italy; mgaffi@scamilloforlanini.rm.it; 4Department of Diagnostic Imaging, Sandro Pertini Hospital, 00157 Rome, Italy; adrenalina_1@hotmail.it; 5Pediatric Emergency Unit, Misericordia Hospital, 58100 Grosseto, Italy; luca.bertacca@uslsudest.toscana.it; 6Unit of Forensic Toxicology (UoFT), Department of Anatomical, Histological, Forensic and Orthopedic Sciences, Sapienza University of Rome, Viale Regina Elena 336, 00185 Rome, Italy; simona.zaami@uniroma1.it; 7Lineberger Comprehensive Cancer Center, University of North Carolina at Chapel Hill, Chapel Hill, NC 27599, USA; aporrelloresearch@yahoo.com; 8Italian National Research Council—Institute of Molecular Biology and Pathology, 00185 Rome, Italy

**Keywords:** RNY1, RNY3, RNY4, RNY5, RO60, DNA replication, cancer microenvironment, cancer etiology

## Abstract

Y RNA are a class of small non-coding RNA that are largely conserved. Although their discovery was almost 40 years ago, their function is still under investigation. This is evident in cancer biology, where their role was first studied just a dozen years ago. Since then, only a few contributions were published, mostly scattered across different tumor types and, in some cases, also suffering from methodological limitations. Nonetheless, these sparse data may be used to make some estimations and suggest routes to better understand the role of Y RNA in cancer formation and characterization. Here we summarize the current knowledge about Y RNA in multiple types of cancer, also including a paragraph about tumors that might be included in this list in the future, if more evidence becomes available. The picture arising indicates that Y RNA might be useful in tumor characterization, also relying on non-invasive methods, such as the analysis of the content of extracellular vesicles (EV) that are retrieved from blood plasma and other bodily fluids. Due to the established role of Y RNA in DNA replication, it is possible to hypothesize their therapeutic targeting to inhibit cell proliferation in oncological patients.

## 1. Introduction

The discovery of Y RNA goes back to 1981 [1], when Lerner and collaborators isolated them in patients affected by systemic lupus erythematosus using specific autoantibodies; this finding was subsequently confirmed in other autoimmune pathologies. The identified targets in these diseases include the soluble ribonucleoproteins (RNP) RO60 (also known as SSA or TROVE2—TROVE domain family, member 2) [2,3] and SSB (small RNA-binding exonuclease protection factor—also known as La) [4]. RNP are complex molecules that include both proteins and RNAs; in RO60 RNP, the RNA component is represented by Y RNA [1,5] which are small non-coding RNAs (sncRNA) that, like other sncRNA, are transcribed by RNA Polymerase III (Pol III) [5,6]. In humans, four genes (*RNY1, RNY3, RNY4,* and *RNY5*) encode for Y RNA; they are clustered on chromosome 7q36.1 [7,8] and their transcripts are named hY1 (112 nucleotides (nt)), hY3 (101 nt), hY4 (93 nt), and hY5 (83 nt), respectively.

The Y RNA family is not limited to the transcripts of the four canonical genes described above; there are other 966 hY RNA pseudogenes (368 for hY1, 442 for hY3, 148 for hY4, and 8 for hY5) scattered on all human chromosomes [9], with at least 878 predicted transcripts. Their distribution is generally proportional to the chromosome length, with the notable exceptions of chromosomes 1, 12, and 17 (excess) and Y (only one pseudogene, possibly because of its peculiar structure and content [10]). Interestingly, no pseudogene sequence is 100% identical to the corresponding hY functional RNA and, notably, although most sequence changes are randomly distributed along Y RNA entire sequence, there is a specific enrichment at specific positions [9], suggesting that these changes are not random, at least in some sequences. We describe below that some of these pseudogenes are indeed transcribed and might fulfill specific functions in tumor biology (see Section 3 about cancer types).

Y RNA are conserved molecules [11,12,13,14,15] (Figure 1) and, in vertebrates, also their clustering is conserved [15,16,17]. A BLAST search shows that the sequence identity is higher than 90% in most vertebrates. Instead, in some non-vertebrate organisms and microorganisms, the sequence similarity with the vertebrate Y RNA is only partial [18] and in plants and fungi they have not been identified yet, thus their evolution is still debated.

## 2. Y RNA Structure and Function

### 2.1. Y RNA Structure

Despite their relatively short length, Y RNA have a complex 3-D structure, including both double-helix rigid stems and single-strand flexible loops [19]; their structure can be roughly split into five major regions, represented by different colors in Figure 2. These regions are responsible for the binding activity of Y RNA with proteins, such as the above mentioned RO60 (which binds the lower stem and the bulge) and SSB (which recognizes the polyU tail). It was shown that the tail may be a variable portion of Y RNA, at least in hY1 and hY3, and that this might influence their intracellular stability, it being a target of exonucleases [20]. The upper stem domain is important for DNA replication, while the loop domain performs different tasks, such as modulation of chromatin association, protein binding, and cleavage for the formation of YsRNA (Y RNA-derived small RNAs) [19,21,22].

### 2.2. Y RNA Interacting Proteins

To date, several proteins have been identified that directly interact with either the entire Y RNA or Y RNA-derived fragments; the current knowledge about these interactions is summarized in Table 1, while the main steps in the life cycle of Y RNA are schematically depicted in Figure 3. It is likely that each Y RNA binds at the same time at least two proteins, one of which is a ‘core protein’ bound on the stem domain or the poly-U tail (such as RO60 or SSB) and another on the loop domain. Indeed, experiments using gel filtration show that Y-linked RNP range from 150 to 550 kDa (see [23] and references therein); this ample size variation likely reflects an equally ample variability in their composition. As shown in Table 1, the enrichment of RNA-processing/stabilizing proteins, including both sncRNA and mRNA is evident; this suggests that Y RNA have, potentially, multiple functions inside the cells, including the control of gene expression. This makes them promising potential targets for cancer therapy.

### 2.3. Role of Y RNA in RO60 Function

A special consideration, among Y RNA interacting proteins, should be given to RO60. This protein is a ring-shaped polypeptide [42] that specifically uses Y RNA as a scaffolding element [43]. Once assembled, this RNP complex fulfills several intracellular tasks, such as RNA quality control [32,44], intracellular transport of RNA-binding proteins [45], and response to environmental stress (reviewed in [22,24]). RO60 is highly conserved [44,46], and its functions include the binding of aberrant or mis-folded non-coding RNAs (ncRNA) such as 5S rRNA or U2 snRNA [47,48]. Due to the very high binding affinity of Y RNA for RO60, some authors hypothesize that Y RNA might act as RO60 repressors [42,49], although some evidence supports the hypothesis that these molecules (or, at least, hY5) might also enhance the recognition of mis-folded ncRNA [32]. Data show that the assembly of Ro RNP protects the particle itself from degradation in several organisms [50]. In addition, RO60 intracellular localization (nucleus/cytoplasm) is also driven by its binding to Y RNA [38,44], possibly through its interaction with other proteins such as nucleolin (NCL), polypyrimidine tract-binding proteins (PTB), and Z-DNA binding protein 1 (ZBP1) [23].

### 2.4. Role of Y RNA in DNA Replication

The role of—at least some—Y RNA in the initiation of DNA replication and cell cycle progression is well established. RNA-mediated depletion (RNAi) has been successfully used to knock down the intracellular amount of hY1, hY3, and hY4, demonstrating that this treatment on any of those is sufficient to halt or strongly reduce both processes, while the artificial re-expression of any of them in the same cells is sufficient to restore a pre-treatment situation [51,52,53,54]. Similar results were achieved by Y RNA inactivation mediated by antisense morpholino oligonucleotides (MOs) micro-injected in vertebrate and worm embryos, causing their death [53,55]. The role of Y RNA in DNA replication is uncoupled from that of RO60 RNP; immuno-depletion in human cells of either RO60 or SSB is not sufficient to inhibit DNA replication [56] and the same happens in case of mutations deleting either the RO60 or SSB binding site inside Y RNA [18,57]. The additional finding that this process is driven by the upper stem domain of Y RNA [18] further supports the fact that Y RNA accomplish DNA replication independently of both RO60 and SSB, which have different binding sites; moreover, it suggests that the initiation of DNA replication depends on different, yet currently unknown, binding proteins [57].

The different results obtained for hY5 (no effect on DNA replication upon RNAi treatment) may be explained in at least two ways. Some authors hypothesize that this Y RNA is just refractory to RNA-mediated depletion [51,52]. Alternatively, hY5 might play a marginal role in this process; this is suggested by its different intranuclear localization: hY1, hY3, and hY4 co-localize on early-replicating euchromatin, while hY5 is mostly localized inside nucleoli [58].

The role of Y RNA in promoting the initiation of DNA replication is in good agreement with their overexpression in various human solid tumors [52] (see also Section 3).

### 2.5. Y RNA Derivatives and Fragments

During the apoptosis, the RNA component of Ro RNP is partly degraded, generating the Y RNA-derived small RNAs (YsRNA); however, Dicer is not involved in their formation, thus their origin and function is not related to those of micro-RNAs [59,60]. These shorter fragments are specifically, abundantly, and rapidly generated from all four Y RNA through the action of caspases [61], yet their causal role in these phenomena (apoptosis and miR biogenesis), if any, is currently unclear [61,62]. These fragments remain bound to the RO60 protein and, in part, also to the SSB protein [61] suggesting their formation occurs early during the apoptotic process. YsRNA have been identified both in healthy tissues [33,59] and in cancer cells. These fragments—especially those derived from hY4—are particularly abundant in plasma, serum [63,64,65], and other biofluids [66,67], where they circulate either as free complexes with a mass between 100 and 300 kDa, or in exosomes and microvesicles [63], collectively called ‘extracellular vesicles’ (EV). Some authors suggest that Y RNA and their derivatives might also fulfill a signaling [63] or a gene regulation [68] function and act also on distant targets.

## 3. Y RNA and Human Cancer

### 3.1. Y RNA Expression is Altered in Human Cancers

The first comprehensive report about Y RNA quantification in human cancers was published by Christov and collaborators in 2008 [52] and, to date, it is still a major reference in this field. The quantification was made by quantitative RT-PCR in extracts from human solid tumors, corresponding nonmalignant normal tissues and derived cultured cells. The examined solid tumors were carcinomas and adenocarcinomas of the lung, kidney, bladder, prostate, colon, and cervix. As a general rule, the authors showed that all four Y RNA are overexpressed, with a range between 4- and 13-fold (for hY4 and hY1, respectively) [52]. Despite its importance, the results of this work have been challenged by some authors in the last years, especially as for bladder and prostate cancers, while in other cases the results were only partially replicated (see for details the following sections about specific tumors). The major points of contrast are the following: (i) as a proliferation biomarker, the authors used antibodies targeting the Ki-67 protein (encoded by the *MKI67* gene), a 359-kD nuclear protein commonly used to detect and quantify proliferating cells, with increased expression associated with cell growth and absent only during the G0 phase of the cell cycle, i.e., in mitotically quiescent cells. Instead, expression levels for each of the four hY RNA were normalized to HPRT1 mRNA; *HPRT1* gene encodes hypoxanthine phosphoribosyltransferase and is involved in the generation of purine nucleotides through the purine salvage pathway. *HPRT1* shows very low variation in expression levels between different human tissues and cell types. The choice of these genes as a reference has been questioned later in works on prostate and bladder cancers (see below) and, for this reason, some authors hypothesize that the results obtained by Christov and co-workers might not be fully reliable, at least in those two tumors. (ii) Christov and collaborators do not distinguish different subtypes of cancer samples, and this might explain the only partial overlap of the results on Y RNA expression, for example, in kidney or lung cancers. Indeed, it is possible to identify specific Y RNA signatures in different cell types, thus this is not a trivial point. (iii) The numbers of tumor and control samples are low and, despite the statistical analysis, the possibility to introduce errors is high. In particular, the specimen sizes were as follows, where the first number in parentheses indicates the number of samples of normal tissue and the second indicates the number of samples of cancer tissue: bladder (4;4); cervix (4;4), colon (4;8); kidney (4;15); lung (4;6); prostate (4;5). (iv) In this work, there is no distinction between the intracellular and extracellular—either free or embedded inside EV—amount of Y RNA. Additionally, this point has been discussed in subsequent works, indicating that sometimes differences of Y RNA expression in these two environments are significant and might underline a specific excretion mechanism of these molecules in some cancer types (see also notes in Table 2).

However, there are some points that have been lately confirmed by other studies, thus supporting the validity of this study [52], at least as a pilot. First, in all tumors analyzed and reported below, Y RNA expression is altered. Surely, it is not always an overexpression. Yet, in some way, these molecules ‘respond’ to abnormal cell proliferation, changing their relative abundance either inside cancer cells or as excreted in blood serum, EV, or both. Second, expression levels of Y RNA vary with tissue type, and there are patterns of mis-expression that are tissue-specific. Third, while the expression of hY1, hY3, and hY4 RNA are to some extent linked, the expression of hY5 RNA is somehow unlinked to the others, at least in some cancer types. This might partly reflect the different spatial localization of the Y RNA, with hY5 mainly localized inside the nucleus and the others mostly inside cytoplasm [29,87]. Additionally, this differential expression has been partially validated later in some cancer types and adds greater variability to the possible combinations of over- and under-expressed Y RNA characterizing cancer and non-cancer cells. Fourth, the treatment of cancer cells with siRNA targeting hY1 RNA caused a 2–3-fold reduction of proliferation in EJ30 bladder carcinoma, DU145 prostate carcinoma, ME180 cervical carcinoma, and WI38 lung fibroblast cells.

In 2010 Meiri and collaborators further analyzed the sncRNA profile of several solid tumor types [62]. In addition, this work suffers from evident limitations, the most important being the low sample number (in many cases this number is not reported), the non-fresh sources used for RNA quantification (23 human formalin-fixed paraffin-embedded (FFPE) samples), and the poor characterization of tumor histology (some samples, like lung, are a mix of various tumor types). Consequently, the study of Meiri et al. [62] should be considered a pilot study, similarly to [52]. The microarray analysis revealed several deregulated RNAs, including two Y RNA-derived fragments [62]. One was a 25 nt RNA derived from hY3, while the other was derived from hairpin-folded hY1. In particular, hY3 was upregulated in colon, bladder, breast, lung, prostate, and pancreas tumors, while hY1 was upregulated in bladder, breast, lung, liver, ovary, cervix, pancreas cancer, and, less markedly, in esophagus cancer.

Below we report a brief summary of what is known about Y RNA and specific cancers. Data on the expression differences are summarized in Table 2 and discussed in each of the following subsections.

### 3.2. Y RNA and Bladder Cancer (BC)

The relationships between non-coding RNA, even intended as causal factors, and bladder cancer are complex, and involve, beside Y RNA, long non-coding RNA (lncRNA), circular RNA (circRNA), small interfering RNA (siRNA), Piwi-interacting RNA (piRNA), small nucleolar RNA (snoRNA), etc. [88]. In a first attempt to characterize Y RNA expression in BC, Christov and collaborators [52] found that two human Y RNA, namely hY1 and hY3, are highly overexpressed in BC. Instead, the increase of hY5 is less significant, and hY4 level is comparable to control, healthy tissues. These results were replicated for hY1 and hY3 two years later [62].

In 2017 Tolkach and collaborators reported that all Y RNA are downregulated in BC (mean expression levels being 2- to 4-fold lower than in normal tissue) [69], with *RNY1*, *RNY3*, and *RNY4* expression being highly correlated to each other, whereas *RNY5* expression levels are less distinctly correlated with the other three. These authors noted that the low abundance of hY1, hY3, and hY4 is typical of muscle-invasive BC (MIBC) compared to non-muscle-invasive BC (NMIBC), whereas hY5 levels in those BCs were comparable. Moreover, the low amount of hY1, hY3, and hY4 also correlates with lymph node metastases and advanced grade and, consequently, with patients’ overall (hY1, hY3, hY4) and cancer-specific (hY1, hY3) survival in a univariate (but not multivariate) analysis. No correlation was found with age or gender. The striking difference found in the expression level of Y RNA between the two available studies on BC might be explained in at least three ways. First, by the much lower number of samples (*n* = 4 vs. *n* = 88) that might have limited the reliability of the results shown in the first report. Second, by the different reference gene used in the earlier work, here substituted by the small nucleolar RNA *SNORD43* and the U6 snRNA *RNU6-2*, which are considered suitable reference genes for urological malignancies. Third, by the relative abundance of specific cell cycle stages in the samples, since Y RNA are more abundant during DNA replication and less during mitosis (see above). An increased cell proliferation might suggest an increased Y RNA expression for promoting DNA replication in cancer cells. However, and quite interestingly, a reduced Y RNA expression has been found also in other cancer types (for example in breast and head/neck cancers, see the specific sections). On the other hand, some limitations affect the work of Tolkach and collaborators as well, such as sample age, RNA integrity, or minor contaminations during sample preparation. In conclusion, further studies are needed to assess with higher confidence if, in BC, Y RNA are downregulated, as the more recent study suggests.

### 3.3. Y RNA and Breast Cancer (BrC)

In 2010 Meiri and collaborators found the upregulation of hY1 and hY3 in this cancer [62]. After that, it has been shown that Y RNA fragments can be detected inside the blood serum and plasma [63]. Dhahbi and collaborators used this approach in 2014 to verify if their abundance can be related to breast cancer physiology [73]. They compared the sera of breast cancer cases (*n* = 5) and normal controls (*n* = 5) for the presence and relative abundance of these molecules, finding that a significant proportion (38%) of the identified RNA fragments are indeed Y RNA derivatives (coming from both genes and pseudogenes), with the following subdivision: a major population of 30–33 nt fragments mostly derived from their 5′ end, and a minor population of 25–29 nt fragments almost exclusively derived from their 3′ end. Interestingly, they showed that in breast cancer patients there is both an increase and a decrease of these fragments, but the increase involves mostly 3′ fragments (19/20) and the decrease involves only 5′ fragments (5/5); this occurs despite the fact that 5′ fragments are, in absolute terms, much more abundant in blood serum than their 3′ counterparts (ratio: ca. 20:1). To further validate these data, the same approach was used on 42 additional datasets available from external sources, which revealed a proportion of Y RNA fragments of 13% of the total RNA. Additionally, in these datasets, specific Y RNA fragments either increased or decreased their abundance compared to controls, depending on the BrC diagnosis or tumor characteristics. This result is in good agreement with a previous one describing an enrichment of 3′-end fragments—derived from human hY5—detected in MCF 7 (mammary adenocarcinoma) cells [59].

More recently [75], a study on triple-negative breast cancer (TNBC) showed an increase of expression of *RNY1*, *RNY5* and, above all, *RNY4* expression. In addition, Tosar and collaborators showed that 5’ hY4-derived fragments of 31–33 nt are greatly and significantly enriched in the extracellular space of cultured breast cancer cells, suggesting that such fragments are specifically excreted by these cells [74]. Similar results have been obtained in lung cancer and leukemia (see specific sections). Altogether, these data do not allow establishing a causality, yet suggest that these molecules might be valid biomarkers also in BrC diagnosis.

### 3.4. Y RNA and Brain Cancer

Glioma is the most common tumor of the brain, comprising about 30% of all brain tumors and 80% of all malignant brain tumors. To date, only one paper was published on glioma and Y RNA expression [72]. In this work, the authors analyzed the ncRNA content inside cells, in EV (including micro-vesicles and exosomes) and as free RNP; as for the cell type, the choice was for low-passage patient-derived tumorigenic glioblastoma multiforme (GBM) cells, the most therapy-resistant stem-like cell population and considered the core cell type within the tumor, while the controls were obtained through the transcriptome analysis of primary human and mouse cells of the brain microenvironment, including neurons, astrocytes, endothelial cells, and microglia. The results show that all extracellular fractions, and especially non-vesicular RNPs, are highly enriched in specific 5′ Y RNA fragments of ≈ 32 nt in length. Additionally, in glioma, Y RNA is present in the form of fragments of ca. 32 nt in length, present mainly inside exosomes or in free RNP complexes for hY1, hY4, and hY5 and with no specific differences for hY3 (which also shows the higher amount in all analyzed compartments). In particular, the mapping coverage of Y RNA reads indicates that these molecules are precisely processed, with the cut site mapping inside the loop domain in all cases. The data available do not allow assigning any function to these fragments, but there is increasing evidence of a role of EV as biological regulators in brain tumor progression (reviewed in [89]); this provides a valid basis to hypothesize a Y RNA role, at least as a specific biomarker, in glioma too.

### 3.5. Y RNA and Cervix Cancer

To date, two reports have been published about the Y RNA quantification in cervix cancer [52,62], both showing an increase of Y RNA expression, especially for hY1, hY3, and hY4. Nicolas and coworkers [59] also showed that, in HeLa cells (an immortal cell line derived from cervical cancer cells), there is an enrichment of hY5 fragments upon poly(I:C) treatment. Further studies are needed to validate these data and to verify if, also in this case, the extracellular content of Y RNA is different from the intracellular one. In [52] the authors also show the effects of RNA-interference on HeLa cells. The experiment was successful for hY1 and hY3, where the authors obtained a 2–8-fold reduction of the target RNA, which in turn caused a 2–3-fold reduction of S phase cells in the treated population. Interestingly, this worked for both hY1 and hY3 RNA, but their contemporary inhibition by double RNAi did not result in a synergistic effect, suggesting a redundant role of at least these two Y RNA in DNA replication inside these cells.

### 3.6. Y RNA and Colon Cancer

Even in the case of colon cancer, the work of Christov and collaborators is still the main reference [52]. It shows that in these cells, all Y RNA are upregulated, with hY1 and hY3 being clearly more abundant than in controls and hY4 and hY5 slightly less abundant but still statistically significant. We will not further discuss this study, yet it is worth noticing that these results have been partially confirmed in three other works. In 2010 Meiri and collaborators showed an increase of hY1 fragments (but no increase for hY3) [62]. Nicolas and collaborators described in 2012 an increase of hY5 fragments in HCT 116 (colorectal carcinoma) cells upon poly(I:C) treatment [59]. In 2017, Mjelle and collaborators showed that hY4 is highly expressed in multiple variants, the most frequent containing 32 nt, in blood serum of rectal cancer patients [76]. Interestingly, hY4 is the most abundant among all ncRNA identified, in all the 96 samples analyzed [76]. Altogether, these results point to Y RNA overexpression in colon cancers, although different molecular species were identified in different studies. These differences might be—at least partly—explained in two ways. First, the aim of the work, which not always investigates the role of Y RNA in colon cancer biology (for example, Nicolas and collaborators studied the dependence of Y RNA formation on the miRNA pathway); second, the different methodological approach (quantitative RT-PCR in [52]; high throughput sequencing, microarray and RT-PCR in stored samples in [62]; Northern blotting analysis in [59]; high throughput sequencing (HTS) in fresh samples in [76]) which, in turn, might have influenced the samples isolation method and, consequently, their content.

### 3.7. Y RNA and Head/Neck Cancers

In 2015 Martinez and collaborators reported that Y RNA-derived small RNAs are significantly deregulated in the sera of head and neck squamous cell carcinoma (HNSCC) patients [77]. These patients had, among the others, an enrichment of 30–33 nt fragments deriving from Y RNA (including putative pseudogenes) degradation, and the proportion of these fragments either significantly increased (2/21) or decreased (19/21) for specific species, suggesting a remodeling of the sncRNA networks in HNSCC. Interestingly, also the fragment length and position (either 3′- or 5′-derived) vary in a specific manner. Remarkably, these data resemble what has been described before for breast cancer specimens (see above) and bring the authors to hypothesize that ‘there may be a specific association between at least these two types of cancer (BrC and HNSCC, ed.) and the minor population of circulating small RNAs derived from the 3′ end of YRNAs’ [77], suggesting an important mechanistic role in cancer biology for these molecules. Also in this case, the role of Y RNA in the tumorigenesis is not clear.

A recent work from Dhahbi and coworkers explored the presence of Y RNA fragments in oral squamous cell carcinoma (OSCC), a form of HNSCC and the most common type of head and neck cancer [78]. In addition, in this case, the authors found that multiple 5′-end Y RNA fragments displayed significantly lower expression levels in the circulation and/or tumor tissue, as compared to their control counterparts. The low number of identified Y RNA in this report might explain the fact that no increase has been detected, as in the previously cited manuscript. Hence, circulating Y RNA seem to specifically characterize this type of tumor.

### 3.8. Y RNA and Blood Cancers

Bernatsky and collaborators demonstrated that many cancer types, and hematologic malignancies in particular, are substantially increased in patients affected by systemic lupus erythematosus [90]. Chakrabortty and coworkers [70] showed that cancer-derived EV (K562 cells, myelogenous leukemia) are enriched for hY5, and that this is the most abundant RNA after rRNA and tRNA. This RNA can be found either complete or as 5′-end fragments of 23, 29, and 31 nt or as a 31 nt fragment of the 3′ end. Interestingly, the complete RNA is present in both cells and EV, while fragments (especially the 5′-end longer forms of 29 and 31 nt) are typical of, and abundant in EV; moreover, these fragments are likely produced inside the EV, and not inside the cell, before EV release in the surrounding microenvironment. In addition, treating primary cell cultures with, or the ectopic overexpression of, the 31 nt-processed fragments of hY5 is sufficient to induce apoptosis, in a dose-dependent manner, in primary cells of multiple developmental origins. Instead, this treatment is inefficient on cancer cells, suggesting that they are somehow insensitive to this treatment. A possible cause of this response is the ability of either EV or of a synthetic version of the 31-nt form of hY5 to change the gene expression profile of several hundred genes in at least two human primary cell lines (BJ, normal skin fibroblasts, and HUVEC, normal human umbilical vein endothelial cell). The comparisons of the effects on gene deregulation of EV vs. hY5 synthetic fragments, and of BJ vs. HUVEC response, allowed the identification of 141 commonly differentially expressed genes with a significant enrichment for pathways related to G2/M DNA replication checkpoints, proliferation, response to replication stress, ERBB2 signaling, and apoptosis. Notably, the authors also show that the apoptosis can be induced even in the absence of direct physical contact between cancer and normal cells [70]. They also hypothesize that this might be a way for cancer cells to create a microenvironment favorable for their own growth, promoting their own invasion and inflammation [70].

More recently, Haderk and collaborators studied the role of tumor cell-derived exosomes in the crosstalk with monocytes in patients affected by chronic lymphocytic leukemia (CLL) [71]. They found that hY4 is highly enriched in exosomes from plasma of CLL patients compared with healthy donor samples, a situation partly resembling the data of Tosar and collaborators for BrC [74]. Moreover, either exosomes or hY4 alone is sufficient to activate cytokine release in monocytes and trigger in these cells the activation of Toll-like receptor 7 (TLR7) [71], a protein that specifically recognizes single stranded RNA and participates in the activation of the innate immunity. Interestingly, the pharmacologic inhibition of endosomal TLR attenuates CLL development in vivo, linking Y RNA to cancer-related inflammation [71]. Moreover, this work also highlights that, in CLL patients, the PD-L1 pathway is activated, allowing tumor cells to escape the immune response, thus linking Y RNA expression to cancer cell survival through a tumor-supportive microenvironment.

Notably, both these works suggest that sncRNA-loaded EV play a pivotal role in leukemia cell survival through the creation of a favorable microenvironment. Moreover, they also show that Y RNA (together with other sncRNA) might potentially act on physically distant, normal cells using EV as shuttles, similar to what has been proposed also for Kaposi’s sarcoma (see below).

### 3.9. Y RNA and Kidney Cancer

In 2008, Christov and collaborators found Y RNA overexpression in their kidney cancer samples, especially for hY1, hY3, and hY4 [52], but in 2010 Meiri and collaborators did not describe any Y RNA overexpression of hY1 and hY3 [62]. However, in 2016 Nientiedt and collaborators specifically investigated the Y RNA profile of clear cell renal cell carcinoma (ccRCC) patients [79]. They analyzed two cohorts of patients; the first (screening cohort; 30 ccRCC and 15 normal renal tissues) was used to quantify all four Y RNA. On this basis, they analyzed a second, independent cohort (validation cohort; 88 ccRCC and 59 normal renal tissues) and found that the expression of *RNY3* and *RNY4* is significantly increased in ccRCC samples; moreover, the expression levels of *RNY4* alone is inversely correlated with the ccRCC stage and the presence of lymph node metastases (though, this last result needs further validation). Nonetheless, the study failed to find clear differences in the blood serum of patients vs. controls for any Y RNA; this implies that in this case, Y RNA could be potential tumor markers only when directly using cancer samples. These data are in overall good agreement with those reported before [52] for hY3, hY4 and, partially, for hY5 (a minor increase in the older study; no increase in the more recent one), but with one major difference regarding the level of hY1 (very high in [52], normal in [79]). However, these two studies are different in at least two parameters: the sample size (much smaller in [52]) and the type of RCC, whose subtype is not reported in the older work. These differences might account for the discrepancy about hY1 amount in RCC, suggesting that more studies are required to validate the presence and define the role of Y RNA in kidney cancer.

### 3.10. Y RNA and Pediatric Lymphoma

Anaplastic large cell lymphoma (ALCL) is a fast-growing non-Hodgkin lymphoma (NHL) and accounts for 10–15% of pediatric and adolescent NHL; ALCL in pediatric patients is almost always ALK-positive and mutant cells show the *t*(2;5)(p23;q35) translocation, constitutively expressing the NPM-ALK fusion protein [91]. A recent work of Lovisa and collaborators compared the ncRNA content of EV of 20 pediatric patients with ALCL, five ALCL cell lines, and five controls [80]. The authors, using RNA-seq, found a specific and abundant enrichment (5x) inside EV of fragments derived from the *RNY4* gene, and the highly similar pseudogenes *RNY4P7*, *RNY4P10,* and *RNY4P20*. The most abundant fragment includes the first 32 nt at the 5′-end of *RNY4* transcript and includes the entire stem region and part of the loop. The result was not confirmed by qRT-PCR for the fragment, but was validated for the full length hY4. In these patients, the fragment can be found both as free RNA and inside exosomes, while the full length hY4 is specifically enriched in exosomes of ALCL patients; the upregulation of full length hY4 has been further confirmed in an extended cohort of 44 ALCL samples and 19 controls. The extended cohort was also used to correlate hY4 presence and disease aggressiveness, revealing that the full length hY4 was more abundant in exosomes of ALCL patients with advanced disease stages (stage 3–4 vs. stage 1–2); among these patients, those relapsing had a higher amount of this RNA compared to those with stable, complete remission. Taken together, these data support the quantification of hY4 as a potential biomarker for both tumor identification and staging, through a non-invasive analysis (liquid biopsy), an important aspect in children management.

### 3.11. Y RNA and Lung Cancer

Lung cancer (LC) is the malignancy with the highest incidence for both sexes worldwide. Data available for this type of cancer and Y RNA expression come from three studies. The first is the abovementioned work of Christov et al. [52], which revealed that hY1, hY3 and, above all, hY5 are overexpressed in this cancer, while hY4 does not show any significant difference when compared to controls. The same report also shows that RNAi against hY1 is sufficient to reduce the number of S phase WI38 lung fibroblast cells, likely by inhibiting DNA replication. The work of Meiri and co-workers [62] confirmed the data about the expression of hY1 and hY3 in this tumor.

Recently, one additional report specifically investigated the link between Y RNA and LC [81]. In this research, Li and collaborators explored the expression of small RNAs in plasma EV from lung adenocarcinoma (ADC) patients, lung squamous cell carcinoma (SQCC) patients, and healthy controls. ADC and SQCC are the most common types of non-small cell lung cancer (NSCLC), which comprises ca. 85–90% of all LC tumors worldwide [92]. The authors found that, among the others, 5′ hY4 pseudogene-derived fragments have a significantly higher expression in plasma EV of NSCLC patients, suggesting its role as a potential biomarker. Moreover, they described that (i) hY4-derived fragments are downregulated inside NSCLS cells and (ii) in cell viability assays, the overexpression of hY4 inhibits the proliferation of the LC cell line A549, suggesting a role of tumor suppressors for the hY4-like fragments in this cancer. This would explain why this Y RNA was not found overexpressed in the samples analyzed before [52]. Thus, the authors hypothesize that the LC cells, to proliferate, selectively excrete hY4 fragments (and other ncRNA) in the plasma in the form of EV.

### 3.12. Y RNA and Skin Cancer

Lunavat and co-workers analyzed the ncRNA content of EV of the melanoma cell line MML-1 [83]. Additionally, in this case, it was possible to highlight a specific signature as for Y RNA present in EV, with hY1, hY4, and hY5 significantly more abundant in all EV (including apoptotic bodies, microvesicles, and exosomes) than inside melanoma cells, while hY3 seemed slightly more abundant inside MML-1 cells. These data were only partly validated by the recent work of Solé and collaborators [84], who analyzed the circulating transcriptome of plasma from melanoma patients. They found that around 40% of all mapped reads from NGS sequencing were Y RNA in both controls and melanoma cells. In particular, they identified 322 different YRNA and YRNA-associated sequences, belonging to three canonical Y RNA sequences (*RNY1*, *RNY3*, and *RNY4*; average: 26.1% of reads), 30 Y RNA pseudogenes (average: 48.4% of reads), and 194 Rfam predicted Y RNA sequences (average: 25.5% of reads), plus other sequences in traces. The presence of the *RNY4* gene and *RNY4P* pseudogene sequences strongly characterizes these EV, accounting for more than 98% of their respective Y RNA. The authors also related the presence of these Y RNA with the disease stage, finding that five of them (namely: *RNY3P1*, *RNY4P1, RNY4P6, RNY4P18,* and *RNY4P25*) are differentially expressed; *RNY3P1*, *RNY4P1,* and *RNY4P25* fragments show a specific enrichment in stage 0 samples, compared to control samples or stage I/II samples (validation cohort: 22 controls and 58 melanoma patients samples). The discrepancy between the two works might be due to the different source of EV: directly released by melanoma cell cultures in the first, retrieved by plasma of patients in the second. In conclusion, also in melanoma cells, the relative amount of Y RNA can potentially be a good marker of this disease and, possibly, also of its stage.

### 3.13. Y RNA and Prostate Cancer

Prostate cancer is the most common malignancy in men. The work of Christov and collaborators showed that, in prostate cancer, hY1 and hY3 are highly abundant, while hY4 and hY5 levels are comparable to controls [52]. These data had been partly confirmed by the work of Meiri and co-workers, who showed the upregulation of hY3, but not hY1, in the same cancer [62]. Tolkach and collaborators quantified all four Y RNA in prostate cancer samples [82]. The study was performed in archival PCA (prostate adenocarcinoma, *n* = 56), normal (*n* = 36), and benign prostatic hyperplasia (BPH; *n* = 28) samples. Similar to what found for BC (see above), also here all four Y RNA seem to be 2–4-fold lower than in normal tissue. Moreover, the data obtained show that hY4 and hY5 amounts are significantly lower in PCA compared to both control and BPH specimens, while hY1 and hY3 show this reduction as a trend. Interestingly, expression levels of all Y RNA were similar in both BPH and normal prostate tissue. The authors also found that *RNY1*, *RNY3,* and *RNY4* expression levels were highly correlated, while *RNY5* expression was less distinctly correlated with the other three. In addition, they also found that higher *RNY5* expression was associated with poor prognosis measured as biochemical recurrence-free survival. These results are in contrast with those for BC, previously described by the same group [69] (see above), where *RNY1*, *RNY3,* and *RNY4* are highly correlated to the outcome and *RNY5* lacks any association. This finding further supports the use of Y RNA as a biomarker for cancer characterization. Additionally, for prostate cancer, these data do not match those reported before [52,62]. Possible explanations for these differences are described in Section 3.1, and even in [82] there are some experimental limitations, such as the use of macrodissection for sample preparation, which might potentially cause tissue contamination. An independent validation of these results and the analysis of larger cohorts would be advisable to discriminate between these opposite results.

### 3.14. Y RNA and Kaposi’s Sarcoma

Kaposi’s sarcoma (KS) is a multi-organ cancer that can form masses in the skin, lymph nodes, lungs, gastrointestinal tract, and other organs. It is caused by human herpesvirus 8 (HHV8; also called KSHV—Kaposi Sarcoma-associated Herpes Virus) and primarily affects individuals with poor immune function. The endemic form, affecting also young people, and its relation with Y RNA expression, has been recently studied by Ikoma and collaborators [85] in Uganda, where KS is the most common cancer among HIV-infected persons (HIV being also endemic in Uganda). The regulatory activity of sncRNA encoded by HHV8 is performed both in autocrine form, in the expressing cells, and in paracrine form, with sncRNA released from the cells and stably persisting—either packaged in exosomes or loaded into lipoproteins—in circulating bodily fluids like plasma. In their study, the authors compared blood-related sncRNA expression profiles of KS-negative individuals with those with or without HHV8 infection detectable from the oropharynx. The analysis was further extended to patients affected by HIV and/or malaria (*Plasmodium falciparum* being endemic as well, in Uganda). Several changes in the total sncRNA profile were detected according to the patient health status and to the ongoing infection(s). The study revealed that, in plasma exosomes, Y RNA are significantly over-represented in all individuals, accounting on average for 93% of the total mapped reads, independently of the HHV8, HIV, and plasmodium infection status. As in other cancers, only 5’-ends-derived fragments were detected, mostly belonging to hY4 (median value: 96%) and, to a lesser extent, to hY1 (1.8%), hY5 (1.4%), and hY3 (<0.1%). The authors suggest that malaria might induce the release from erythrocytes of exosomes (a known response of these cells to plasmodium, and for which the presence of Y RNA was already described [86]) that contain mainly hY4 fragments; these vesicles would target distant monocytes and activate the TLR7/8 signaling pathway which, in turn, would induce HHV8 reactivation from latency, thus promoting KS [85]. If these data will be confirmed and the model validated, Y RNA might become potential therapeutic targets in KS treatment.

### 3.15. A Possible Role of Y RNA in Other Cancer Types

As reported above, autoimmune diseases cause systemic autoimmune chronic inflammation, also when RO60 is the target. Over the years, evidence has accumulated of a direct relation between tumorigenesis and alterations of the RO60 function, as well as between tumorigenesis and inflammation (see for example the section above, about blood cancers). Although a direct role of Y RNA has not been recognized in these cases, the strong interaction of these molecules (RO60 and Y RNA) might help understand their role in at least some of these pathologies. For these reasons, we briefly report here what is the current knowledge on this topic.

It has been repeatedly shown that patients affected by autoimmune RO60-dependent conditions have a higher frequency of malignancies, including melanoma, T-cell lymphoma, non-Hodgkin lymphoma, and breast carcinoma [90,93,94,95,96]. As for melanoma and non-melanoma skin cancer (NMSC) in these patients, it is not yet clear if RO60 is directly involved, or if the auto-antibodies cross-react with other targets, possibly being part of the same RNP complexes, or even if inflammation per se has a direct, stimulatory role in cell growth, especially for NMSC (see [96] and references therein). This suggests that the role of RO60 RNP might be far more complex than expected in these tumors and that this protein, and in turn the Y RNA it binds, could be also an indirect cause of cancer formation.

Pancreatic ductal adenocarcinoma (PDAC) is one of the most lethal human cancers, with an extremely poor prognosis. The work of Liu and collaborators [97] shows that RO60 expression is increased in PDAC tissues compared with normal pancreatic tissues, while the knockdown of RO60 by siRNA significantly decreases cell proliferation and invasion in vitro and inhibits the growth of subcutaneous tumors in vivo. Even in this case, it would be interesting to explore the possibility to downregulate RO60 by targeting Y RNA.

## 4. Conclusions

Data about Y RNA biology and, more specifically, about their role in cancer etiology, began to accumulate in the last years. Despite the sparse and sometimes contradictory data, a role for these molecules in cancer formation is emerging. The interaction with several proteins involved in gene expression (Table 1), the role of Y RNA in DNA replication, the putative role of some of them as potential tumor suppressors, the possibility to hit distant targets through EV or other carriers (such as free RNP in plasma), and their possible role in establishing a tumor-friendly microenvironment, all point to their possible involvement in tumorigenesis. On the other hand, the differential expression of Y RNA among different tissues, across different tumors and, sometimes, also between tumor cells and their released EV (Table 2) support their possible use in the identification and characterization of tumors, often without the use of invasive biopsies, but through a simple analysis of bodily fluids. However, this potential diagnostic strength is balanced by the complexity of Y RNA quantification demonstrated by the available literature, where several different approaches have been used to address this topic (Table 2). Choosing one (or a limited set of) reference genes as a general tool for all cancers is not recommended in the case of qRT-PCR. Moreover, even when using methods that take advantage of internal controls (as in RNA-seq, where the tumor/control expression ratio is used) the choice should be on a tissue-specific basis and possibly also taking into account both cancer sub-type and staging/grading. For information about cancer staging we refer the readers to the official oncological guidelines, which are regularly updated. While some contradictions and methodological limitations exist, further analysis of these elusive molecules in larger cohorts and the use of robust protocols may shed a new light in our understanding of Y RNA function in human cancers.

## Figures and Tables

**Figure 1 cancers-12-01238-f001:**
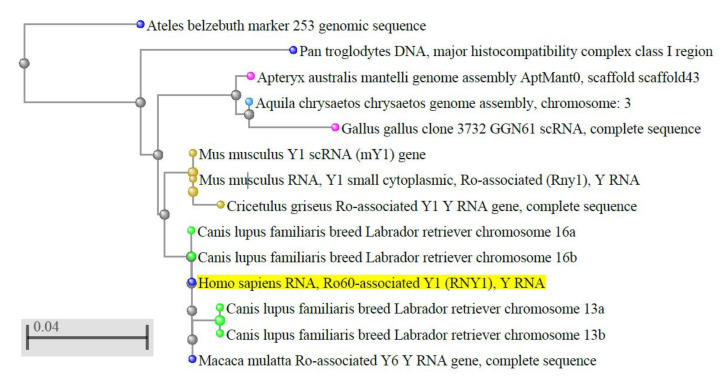
Evolutionary conservation of Y RNA in vertebrates. The dendrogram shows the conservation of human hY1 in monkeys, dogs, rodents, and birds; only best matches (identity >95%) were selected. The tree was obtained through the NCBI BLAST website (URL: https://blast.ncbi.nlm.nih.gov/Blast.cgi, accessed on April 2020), using default settings. Out of the 100 results displayed, we selected only those coming from genome sequences, thus excluding predicted sequences, human pseudogenes, bacterial artificial chromosome (BAC) clones, and other constructs. The color codes, set by default by the NCBI website, are as follows: yellow highlight: query sequence; light blue dots: hawks and eagles; pink dots: birds; brown dots: rodents; blue dots: primates; green dots: carnivores; grey dots: nodes.

**Figure 2 cancers-12-01238-f002:**
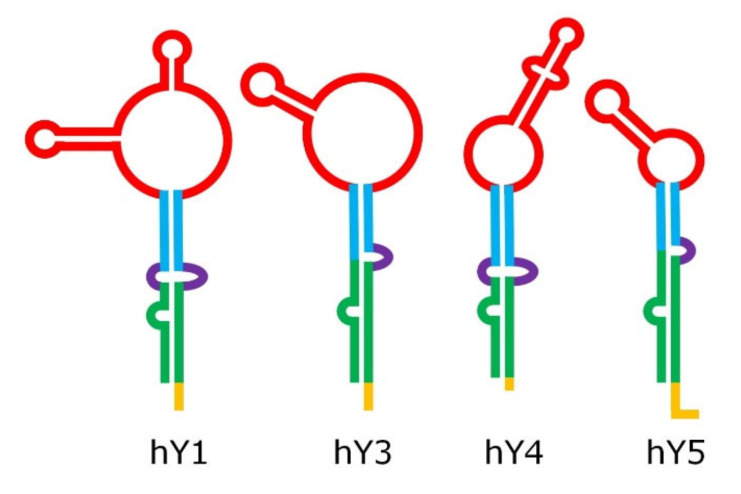
Structure of human Y RNA. The structure was retrieved from the literature [19,21,22], but alternative structures with minor differences have been reported as well [23,24]. The domains are the poly-U tail (yellow), the lower stem (green), the bulge (violet), the upper stem (blue), and the loop (red).

**Figure 3 cancers-12-01238-f003:**
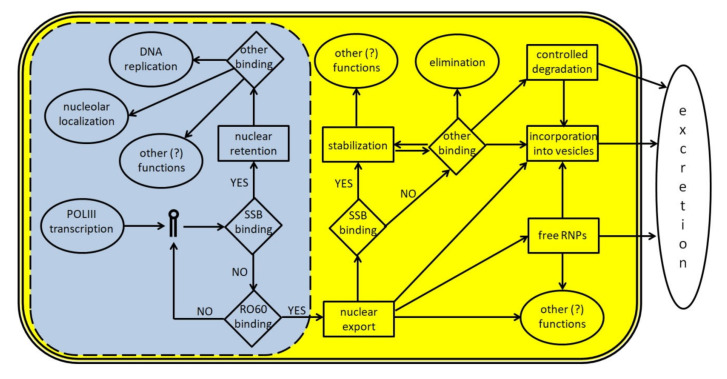
A schematic representation of Y RNA life cycle. The light blue area represents the nucleus (the dotted line indicates the presence of nuclear pores) and the molecular events occurring inside it; the yellow area represents the cytoplasm; the white space represents the surrounding extracellular environment. Y RNA are transcribed by POLIII and, if bound by SSB/La, may remain inside the nucleus to perform specific tasks like promoting DNA replication or other functions, upon binding to specific proteins such as those reported in Table 1. In many cases, these additional functions are not fully understood, since Y RNA binding companions are known, but not their role. If Y RNA are bound by RO60 inside the nucleus, they can be exported into the cytoplasm with the help of specific carrier proteins. Once there, Y RNA may perform several tasks, either alone or in RNP complexes. Y RNA may be stabilized through their binding to SSB, RO60, or other proteins, and they may contribute to the stabilization of several target molecules. Moreover, they may also be excreted in the extracellular environment either as free and complete RNA, or as free RNP complexes, or inside micro vesicles. Y RNA excretion may also occur after a specific cleavage, that generates the YsRNA. Once in the extracellular environment, Y RNA may be internalized by target cells to perform additional tasks.

**Table 1 cancers-12-01238-t001:** Y RNA binding proteins. Data inside parentheses indicate unconfirmed data or minor effects. Protein names (column 1) are those approved by the HUGO (Human Genome Organization) Gene Nomenclature Committee (HGNC). Proteins are listed in alphabetical order according to data in column 1. References (refs) indicate the works that illustrate the protein binding to Y RNA (i.e., not the protein function). Y RNA between parentheses indicate weak or unconfirmed data; 1-3-4-5 is for hY1-hY3-hY4-hY5, respectively.

Protein (HGNC)	Synonym(s)	Interacting Y RNA	Y RNA Domain Involved	Protein Function	Refs
AGO1	EIF2C1, AGO	unknown	unknown	gene silencing through RNAi	[25]
APOBEC3F	ARP8	(1), (3), (4), (5)	unknown	antiviral activity	[26,27]
APOBEC3G	CEM15	1, 3, 4, 5	unknown	antiviral activity	[26,27]
CALR	CR, CRT	1, 3, 4, 5	unknown	formation of the RO60 RNP complex, calcium-binding chaperone	[28]
CPSF1	CPSF160	1, 3	loop	mRNA poly-adenylation	[29]
CPSF2	CPSF100	1, 3	loop	mRNA poly-adenylation	[29]
CPSF4	NEB1	1, 3	loop	histone pre-mRNA processing	[29]
DIS3	EXOSC11	(1), (3)	polyU tail	Y RNA stabilization	[20]
DIS3L	DIS3L1	1, 3	polyU tail	Y RNA degradation and turnover	[20]
EXOSC10	PMSCL2	1, 3, 4, 5	polyU tail	Y RNA trimming, stabilization	[20]
ELAVL1	HuR	3	unknown	mRNA stabilization	[29]
ELAVL4	HuD	3	loop	mRNA stabilization, mRNA translation	[30]
FIP1L1	FIP1-like 1	1, 3	loop	mRNA poly-adenylation	[29]
HNRNPK	HNRPK	1, 3	loop	pre-mRNA binding	[31]
IFIT5	RI58	5	unknown	innate immunity	[32]
MATR3	VCPDM	1, 3	upper and lower stem	nuclear matrix, transcription, RNA-editing	[29,33]
MOV10	gb110, KIAA1631	unknown	unknown	microRNA-guided mRNA cleavage	[34]
NCL	nucleolin, C23	1, 3	loop	association with intranucleolar chromatin	[35]
PARN	DAN	1, 3, 4, 5	polyU tail	Y RNA trimming, stabilization	[20]
PTBP1	hnRNP I, PTB	1, 3	loop	pre-mRNA splicing	[31]
PUF60	RoBPI, FIR	(1), (3), 5	(loop)	pre-mRNA splicing, apoptosis, transcription regulation	[32,36]
RNASEL	PRCA1, RNS4	1, (3), 4, 5	loop	cell cycle arrest and apoptosis	[37]
RO60	TROVE2, SSA	1, 3, 4, 5	lower stem	stabilization, nuclear export, RNA quality control	[38,39,40]
RPL5	L5	5	loop	5S rRNA quality control	[32]
SSB	La, LARP3	1, 3, 4, 5	polyU tail	nuclear localization, protection of 3′ ends of pol-III transcripts	[39]
SYMPK	SYM, SPK	1, 3, (4), (5)	loop	mRNA poly-adenylation, histone pre-mRNA processing	[29]
TENT4B	PAPD5	1, 3, 4, 5	polyU tail	Y RNA oligoadenylation, degradation	[20]
TOE1	PCH7	1, (3)	polyU tail	Y RNA degradation and turnover	[20]
YBX1	NSEP1	1, 3, 4, (5)	unknown	mRNA transcription, splicing, translation, stability	[29,34]
YBX3	DBPA	unknown	unknown	cold-shock domain protein; DNA-binding domain protein	[34]
ZBP1	C20ORF183, IGF2BP1	(1), 3	loop	nuclear export of RO60 and Y3	[34,41]

**Table 2 cancers-12-01238-t002:** Expression levels of Y RNA in various cancer types. Cancers are listed in alphabetical order according to the affected organ, irrespective of their histology, for which we refer the reader to the main text; KS means Kaposi’s sarcoma, a multi-organ cancer. The word ‘serum’ is used for short to indicate blood serum. An arrow pointing upward means overexpression; an arrow pointing downward means under-expression; a horizontal, double-headed arrow indicates no significant change; arrows between parentheses indicate weak evidence. N/A means that no data are available. Refs indicates bibliographic references, while ref gene indicates the gene(s) used for quantitative comparison. See the text for further details.

Cancer	hY1	hY3	hY4	hY5	Refs	Sample Type	Sample Number	Control Number	Method	Ref Gene	Notes
bladder	↑	↑	↔	(↑)	[52]	cell cultures	4	4	qRT-PCR	Ki-67, HPRT1	
	↑	↑	N/A	N/A	[62]	FFPE	5	1	(a)	hsa-miR-200b	
	↓	↓	↓	↓	[69]	FFPE	88	30	qRT-PCR	SNORD43, RNU6-2	
blood	N/A	N/A	N/A	↑	[70]	K562 cells EV	N/A	N/A	RNA-seq	N/A	1
	(↑)	N/A	↑	N/A	[71]	plasma EV	N/A	N/A	RNA-seq	N/A	1
brain	↑	↔	↑	↑	[72]	cell culture EV, free RNP	N/A	N/A	RNA-seq	N/A	
breast	↑	↑	N/A	N/A	[62]	FFPE	5	N/A	(a)	hsa-miR-200b	
	see text	see text	see text	see text	[73]	serum	5	5	RNA-seq	N/A	2
	N/A	N/A	↑	N/A	[74]	cell culture EV, free RNP	N/A	N/A	RNA-seq	N/A	
	↑	N/A	↑	↑	[75]	cell lines	26	N/A	RNA-seq	N/A	
cervix	↑	↑	↑	↑	[52]	cell cultures	4	4	qRT-PCR	Ki-67, HPRT1	
	N/A	N/A	N/A	↑	[59]	HeLa cells	N/A	N/A	northern blotting	N/A	
	↑	(↑)	N/A	N/A	[62]	FFPE	N/A	N/A	(a)	hsa-miR-200b	
colon	↑	↑	↑	↑	[52]	cell cultures	8	4	qRT-PCR	Ki-67, HPRT1	
	N/A	N/A	N/A	↑	[59]	HeLa cells	N/A	N/A	northern blotting	N/A	
	↑	↔	N/A	N/A	[62]	FFPE	N/A	7	(a)	hsa-miR-200b	
	N/A	N/A	↑	N/A	[76]	PE	96	N/A	HTS	miR-128a-3p, miR-92a-3p, miR-151a-3p	
esophagus	(↑)	↔	N/A	N/A	[62]	FFPE	N/A	N/A	(a)	hsa-miR-200b	
head/neck	see text	see text	see text	see text	[77]	serum	N/A	N/A	RNA-seq	N/A	2
	see text	see text	see text	see text	[78]	serum, tumor tissue	5+2	5+2	qRT-PCR	β2-microglobulin	2
kidney	↑	↑	↑	↑	[52]	cell cultures	15	4	qRT-PCR	Ki-67, HPRT1	3
	↔	↔	N/A	N/A	[62]	FFPE	N/A	N/A	(a)	hsa-miR-200b	
	↔	↑	↑	↔	[79]	tissue, serum	30+88	15+59	qRT-PCR	SNORD43	
liver	↑	↔	N/A	N/A	[62]	FFPE	N/A	3	(a)	hsa-miR-200b	
lymphatic system	N/A	N/A	↑	N/A	[80]	fresh, cell lines	20+5+44	5+19	RNA-seq	N/A	
lung	↑	↑	↔	↑	[52]	cell cultures	6	4	qRT-PCR	Ki-67, HPRT1	1
	↑	↑	N/A	N/A	[62]	FFPE	6	4	(a)	hsa-miR-200b	1
	N/A	N/A	↑	N/A	[81]	plasma EV, cell cultures	44+31	17	RNA-seq, qRT-PCR	U6 snRNA	
ovary	↑	↔	N/A	N/A	[62]	FFPE	N/A	N/A	(a)	hsa-miR-200b	
pancreas	↑	↑	N/A	N/A	[62]	FFPE	N/A	N/A	(a)	hsa-miR-200b	
prostate	↑	↑	↔	(↑)	[52]	cell cultures	5	4	qRT-PCR	Ki-67, HPRT1	
	↔	↑	N/A	N/A	[62]	FFPE	N/A	N/A	(a)	hsa-miR-200b	
	↓	↓	↓	↓	[82]	FFPE	56	36+28	qRT-PCR	SNORD43, RNU6-2	
skin	↑	(↑)	↑	↑	[83]	MML-1 cells	N/A	N/A	RNA-seq	N/A	1
	↑	↑	↑	N/A	[84]	plasma EV	118	99	RNA-seq, ddPCR	N/A	1
KS	↑	(↑)	↑	↑	[85]	plasma EV	8+28	19	RNA-seq	N/A	1
	↑	N/A	↑	N/A	[86]	plasma EV	N/A	N/A	RNA-Seq	N/A	1

Notes. FFPE are formalin-fixed paraffin-embedded tumor cells from stored samples of various ages; PE are paraffin-embedded cells from fresh samples. qRT-PCR is quantitative Real Time PCR; HTS is high throughput sequencing. (a): the authors used simultaneously three methods (high throughput sequencing, microarray analysis and qRT-PCR) and compared the obtained results. 1: differences between EV and cancer cells; 2: differential up- and down-regulation (see text); 3: differences between blood serum and cells.

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
