# Peer review of "Y RNA: An Overview of Their Role as Potential Biomarkers and Molecular Targets in Human Cancers"

_cancers, 2020, doi:10.3390/cancers12051238_

Round 1
Reviewer 1 Report
This review summarize recent findings on YRNA in different cancers. The review is well written and comprehensive. I have some comments which might improve the readability for readers, in particular those who are not expertise in the YRNA field.
1) line 254-255: They compared the sera of breast cancer cases (n=5) and normal controls for the presence and relative abundance of these molecules. The number of normal controls should also be indicated.
2) line 306-316: Y RNA and colon cancer
There are several colon cancer studies on Y RNA. Were hY1, hY3, hY4 and hY5 all investigated in those studies? And in case there was inconsistent finding, for example, Christov et al found hY1 and hY3 more abundant in cancer than in control, whereas hY1 but not hY3 was increased in Nicolas et al's study, can the authors discuss the possible reasons?
A table summarizing the number of samples, type of sample, method of detection, reference gene and pattern of Y RNA (increase/decrease/no change) compared to control in studies of different cancer types will be useful for comparison.
3) According to the authors' expertise, what should be the best reference gene for quantifying Y RNA level in tissue and serum/plasma/EV?
4) In some cancers, hYx was studied whereas in some RNYx was mentioned. Is there any difference between them? For example, line 465-469, "the data obtained show that hY4 and hY5 amounts are significantly lower in PCA compared to both control and BPH specimens, while hY1 and hY3 show this reduction as a trend. Interestingly, expression levels of all Y RNA were similar in both BPH and normal prostate tissue. The authors also found that RNY1, RNY3 and RNY4 expression was highly correlated, while RNY5 expression levels were less distinctly correlated to the other three." What is the major difference between hY1 and RNY1, etc? It will be somehow difficult for non-expertise to follow in this area.
5) A minor comment, since in line 217 the authors mentioned the summary of Y RNA in specific cancers was listed in alphabetical order, but the topic of some cancer was not actually following this order, probably due to different name for specific cancer was used in the topic sentence. I think the authors can either re-arrange the order, change the name of cancer in the topic or just simply remove the "alphabetical order" sentence.
Author Response
Dear Editor of Cancers,
we would like to thank your Journal for considering our manuscript worth of revision, and we are grateful to the Reviewers for the very positive comments and the provided suggestions, which we carefully used and significantly increased the manuscript quality and readability.
Below are our point-to-point responses to the Reviewers. We hope that these text modifications will meet the requests. We have also further improved the language quality and style throughout the text.
Best regards,
Roberto Piergentili, on behalf of all Authors.
Responses to Reviewer #1.
- The Reviewer noted that the number of controls in lines 254-255 (breast cancer) is missing.
- This information has been added to the text as requested; these data are also reported in the expanded Table 2 (see also point 2).
- The Reviewer asked to comment the inconsistencies in colon cancer data, and suggests to add a Table including the number of samples, type of sample, method of detection, reference gene and pattern of Y RNA (increase/decrease/no change) compared to the control.
- The study of Christov et al. investigated “specifically” all Y RNA and demonstrated that all of them are over-expressed, although this is more evident for hY1 and hY3. The other three works cited in the paragraph all report an increase in Y RNA expression, but limited to the RNA cited in our report; that’s why we wrote that Christov’s results were “partially confirmed”. In the work of Meiri and coworkers, the authors isolated the population of expressed small RNAs in various tumors and then sequenced them. They reported that only two of them (hY1 and hY3) are abundant and only one of them is over-expressed in colon cancer; the other one is present, but not overexpressed (Fig 5 of Meiri’s work). So, hY4 and hY5 were not investigated at all, because they were not found in the starting pool. Similarly, Nicolas’ work investigated solely hY3 and hY5 and no mention is found in their manuscript of the other two; in addition, the work aim was to investigate Y RNA biogenesis and their relation with Ago, and not Y RNA role in tumorigenesis. Finally, Mjelle and coworkers specifically investigated colon cancer at different stages, but focused on the whole sncRNA population and then isolated the molecules of interest, without further analysis on why some molecules were not present. It is clear that the four works were not aimed at creating a direct link between Y RNA expression and specific cancers, thus in some cases the authors limited their investigations to what they found in the initial pool of sncRNA; for this reason, making a direct comparison between these works is difficult and, we believe, scientifically not fully appropriate. The methodological approach is, indeed, different: Christov et al. used quantitative RT-PCR and a lately questioned reference gene; Meiri et al. used three different approaches (high throughput sequencing, microarray, and RT–PCR); Nicolas et al. used a simple Northern blotting analysis; Mjelle et al. used a high throughput sequencing (HTS) method. These different approaches require different tissue preparations and this might partly explain the differences in the results, although the shared evidence is an overexpression of specific Y RNA. Moreover, some methods use a reference gene (qRT-PCR) while others use internal controls (RNA-seq) or the tumor/control expression ratios. To take this into account, and to add as little text as possible to the manuscript (which already exceeds word limit set by the journal) we added two sentences at the end of paragraph 3.6 to briefly explain all these issues. We hope that the Reviewer finds these sentences sufficient to address his/her request.
- As for the Table, we expanded the data of Table 1 by adding the suggested information.
- The Reviewer asks to comment on the best reference gene for quantifying Y RNA level in tissue and serum/plasma/EV.
- Our work is aimed at giving some different messages to the scientific community working on Y RNA and cancer. One of them is that these molecules have a differential expression pattern, according to the primitive cancer type and, possibly, its sub-type, stage and grade (as in colon cancer). Moreover, these cancers also have different RNA content according to the source of RNA (EV, cell cultures, tissue samples) and, to further complicate this matter, the literature shows a certain degree of contradiction, at least for some cancers. Therefore, selecting one reference gene for all cancers is not possible, and even the choice of tissue-specific genes should be done very carefully, since many works need to be further validated. We believe that the choice should be tissue-specific and, possibly, also taking into account both cancer sub-type and staging/grading, for which we refer the readers to the official oncological guidelines, which are regularly updated. To stress this point, we have slightly expanded the Conclusions section.
- The Reviewer asks the difference in the use of hYx vs. RNYx in the manuscript.
- When we refer to RNYx, we talk about the gene coding for the Y RNA; when we refer to hYx, we refer to the transcribed RNA. This is stated at the beginning of the introduction, lines 53-56. The possible confusion is due to the fact that some reported works are about “gene expression”, while others are about “RNA quantification”; we decided that it was better to stick to the original choice made by the cited authors in their manuscripts. However, we also checked the entire text for consistency and made the following changes: (i) in each case the RNYx (gene) was cited, the italic text was used; (ii) in one case (paragraph 3.8), we mistakenly used the gene name instead of the RNA name; we fixed it. We thank the reviewer to highlight this possible source of confusion.
- The Reviewer asks to fix the issue regarding the "alphabetical order" in line 217.
- We decided that the easiest way to solve this issue was to remove any reference to the use of “alphabetical order”. We thank the reviewer for pointing out this mistake.
Reviewer 2 Report
The review paper entitled "Y RNA: an overview of their role as potential biomarkers and molecular targets in human cancers" discusses (mostly) the expression levels of Y RNA in human cancers emphasizing on potentials of Y RNA overexpression to be used as a biomarker for early cancer detection. Indeed, Y RNAs are a class of small non-coding RNAs that are well demonstrated to have a role in cell cycle progression and tumor growth.
Overall, the paper can be divided into two parts: a general review of the structure and function of Y RNAs, and a more detailed review of previous works about the expression of Y RNAs in various human tumors. I found this structure appropriate and neat. Another advantage of the paper is that the author added their thought and appropriately discussed the literature. In addition, the authors included a wide range of cancer types (bladder, breast, brain, cervix, colon., head and neck, blood, kidney, lymphoma, lung, skin, and prostate) which is fine. There are some points that the author should consider to improve this work:
- In table 2 blank cells can confuse the reader. “N/A” refereeing to “not available” should be added in the blank cells.
- In my opinion, a schematic figure showing the lifecycle and function of Y RNA in the cells should be added to the paper. This will help the reader to better understand Y RNA biology and its difference to other non-coding RNAs.
- To illustrate the conserved domains of Y RNA, the Y RNA sequence from humans and another example organism can be shown by blast and multiple sequence alignment. For this, NCBI Nucleotide Blast web server can be used.
- The author can also show the expression levels of Y RNA in cancers by a bar graph or heatmap, which will be very good and let readers to better understand the expression levels of Y RNA and that which cancer type expresses more or less at a glance. At a higher level, the author can simply analyze the Non-coding RNA profiling datasets from cancer cells to see the expression levels of Y RNA in various tissues. These data can be obtained from GEO database and the analysis does not require high bioinformatic skills. Indeed, this is not necessary, I would like to let the authors know if they are interested in extending this paper.
Author Response
Dear Editor of Cancers,
we would like to thank your Journal for considering our manuscript worth of revision, and we are grateful to the Reviewers for the very positive comments and the provided suggestions, which we carefully used and significantly increased the manuscript quality and readability.
Below are our point-to-point responses to the Reviewers. We hope that these text modifications will meet the requests. We have also further improved the language quality and style throughout the text.
Best regards,
Roberto Piergentili, on behalf of all Authors.
Responses to Reviewer #2.
- The Reviewer suggests filling the blank cells of Table 2 with “N/A” because blank cells might be confusing.
- The Table has been changed as advised; moreover, a new and expanded version of Table 2 is provided, according to the requests of the other Reviewer.
- The Reviewer suggests adding a schematic figure showing the lifecycle and function of Y RNA in the cells, which will help the reader to better understand Y RNA biology and its difference vs. other non-coding RNAs.
- Figure 3 has been added to the manuscript depicting the Y RNA life cycle; this figure uses a flow diagram overlaid on cell compartments.
- The Reviewer suggests adding a figure depicting the conservation of Y RNA sequence from humans and another example organism by blast and multiple sequence alignment, taking advantage for example of the NCBI Nucleotide Blast web server.
- We BLASTed the human hY1 sequence using the suggested on-line tool and obtained a figure depicting the evolutionary relationship of this molecule with some homologues in mammals and birds. This figure is now Figure 1; the former Figure 1 is now Figure 2. We hope we met the Reviewer’s request by generating this figure.
- The Reviewer suggests using the GEO database to create bar graphs or heatmaps to better illustrate the expression levels of Y RNA and which cancer type expresses more or less any hYx at a glance. This request is more a suggestion for expanding the manuscript, as specified by the Reviewer himself/herself.
- We sincerely thank the Reviewer for this suggestion. We discussed this idea and our conclusion is that it would be enough to write a separate scientific manuscript. Indeed, adding only a few lines and a figure to the present review on this topic would likely end up in a very incomplete information. For this reason, we decided not to proceed with this part, which we are considering for a new publication.